# Effect of Sarecycline on the Acne Symptom and Impact Scale and Concerns in Moderate-to-Severe Truncal Acne in Open-Label Pilot Study

**DOI:** 10.3390/antibiotics12010094

**Published:** 2023-01-05

**Authors:** Angela Yen Moore, Kara Hurley, Stephen Andrew Moore, Luke Moore, Ilana Zago

**Affiliations:** 1Arlington Center for Dermatology, Arlington, TX 76011, USA; 2Arlington Research Center, Arlington, TX 76011, USA; 3Department of Dermatology, Baylor University Medical Center, Dallas, TX 75246, USA; 4Department of Medical Education, School of Medicine, Texas Christian University, Fort Worth, TX 76017, USA; 5Texas College of Osteopathic Medicine, University of North Texas Health Science Center, Fort Worth, TX 76107, USA

**Keywords:** acne vulgaris, truncal acne, tetracycline, sarecycline, ASIS, PRO, PROM

## Abstract

Truncal acne is common, and the psychosocial burden may be underestimated as patients most often complain of facial acne. The Acne Symptom and Impact Scale (ASIS) is a 17-item patient-reported outcome (PRO) measure designed to assess the signs and impacts of acne vulgaris. ASIS has previously been validated in a prospective, non-interventional study as a reliable PRO instrument for facial acne. In a pilot study, ASIS, and an additional 10 new questions that focused on the concerns of patients (ASIS-C), were given to 10 patients with moderate-to-severe truncal acne vulgaris who received 3 months of monotherapy with oral sarecycline, a narrow-spectrum tetracycline-class antibiotic. ASIS-C questionnaires were also given to 10 acne-free control subjects. Average ASIS-C answers decreased by 4% for Signs, 15% for Impact, and 16% for Concerns in the 10 patients, with greater decreases of 5% for Signs, 20% for Impact, and 19% for Concerns in the 60% of patients whose truncal acne was clear or almost clear after 12 weeks of sarecycline treatment. In this study, sarecycline was effective in reducing the psychosocial burden associated with truncal acne based on the ASIS-C PRO measures.

## 1. Introduction

Acne vulgaris is a common chronic inflammatory condition that affects adolescents and adults. Most current literature on the pathophysiology and treatment of acne primarily focuses on facial acne. While facial acne is the most common presenting patient complaint, truncal acne may be underestimated [1]. Since the severity of truncal acne does not correlate with the severity of facial acne [2], the psychosocial burden and physical pain and bleeding of truncal acne may also be underestimated [3]. In a study of 696 patients across 5 U.S. cities, >3% of patients presented with only truncal lesions, while ~50% presented with truncal and facial acne [4]. Of the cohort with both truncal and facial acne, ~25% of these patients did not voluntarily mention truncal acne, but clinical examination detected truncal acne [4]. While the majority presented with mild to moderate truncal acne, more than 75% were interested in truncal acne treatment [4].

The Acne Symptom and Impact Scale (ASIS) is a 17-item patient-reported outcome measure (PROM) designed to assess the signs and impacts of acne vulgaris [5]. ASIS has previously been validated in a prospective, non-interventional study as a reliable PROM for facial acne [6]; however, ASIS has not been utilized in evaluating the psychosocial impact of truncal acne. Additionally, 10 new items have been proposed to be added to the ASIS questionnaire to address the patient’s concerns. These items have not yet been validated. We refer to this questionnaire as ASIS-C. The aim of this study is to investigate if there is a role for ASIS-C in the evaluation of the impact of truncal acne.

Sarecycline is a narrow-spectrum tetracycline-class antibiotic approved in October 2018 by the Food and Drug Administration (FDA) for the treatment of moderate-to-severe acne vulgaris in patients 9 years of age and older [7]. Statistically significant improvement was observed with sarecycline in truncal acne, with improvement at week 3 on the back and improvement at week 6 on the chest, based on investigator global assessment (IGA) scores in the pivotal phase 3 trials [8]. Ten patients with moderate-to-severe truncal acne vulgaris, the ASIS questionnaire at baseline, and after 12 weeks of monotherapy with oral sarecycline 1.5 mg/kg/day. We hypothesized that the psychosocial impact of truncal acne may be assessed by measuring ASIS before and after truncal acne treatment.

## 2. Materials and Methods

### 2.1. Study Design

This pilot study is a phase 4, open-label prospective case series conducted at a single dermatology center in the United States. We gave 10 patients with moderate (baseline IGA 3) to severe (baseline IGA 4) truncal acne vulgaris the ASIS questionnaire, along with the new Concerns questions, at baseline and after 12 weeks of monotherapy with oral sarecycline 1.5 mg/kg/day.

### 2.2. Inlcusion Criteria

To be included in this study, patients had to be 9 to 45 years, weigh 33 to 136 kg, and have achieved a score of 3 (moderate) or 4 (severe) on the Investigator’s Global Assessment (IGA) scale for inflammatory lesions of acne of the trunk. An IGA score of 3 is defined as some-to-many non-inflammatory lesions and possibly some inflammatory lesions, but no more than one small nodular lesion [9]. IGA 4 (severe) is defined as some-to-many non-inflammatory and inflammatory lesions, but no more than a few nodular lesions [9].

### 2.3. Exclusion Criteria

Individuals were excluded if they had a dermatologic condition, any chronic illness interfering with study evaluations, any allergy or resistance to tetracyclines, drug-induced acne, hormonal contraceptive initiation, systemic retinoids, systemic corticosteroids, androgens, or anti-androgens within 12 weeks prior to sarecycline initiation.

### 2.4. Control Patients

Ten acne-free control patients were recruited from a local university. A survey was sent out to university students to identify subjects that approximately matched the ages of the treatment subjects.

### 2.5. Methods

The patients treated with sarecycline and the control patients. The validated ASIS questionnaire was divided into Signs and Impact domains. The contents of the ASIS-C questionnaire are shown in Table 1. The patients were evaluated by a single dermatologist at a single dermatology center in the United States. Ten patients with moderate-to-severe acne were given the ASIS-C questionnaire (Table 1). In addition, ten acne-free subjects were given the questionnaire for comparison and matched by approximate age ranges.

### 2.6. Data Analysis

This is a qualitative study. In order to analyze the results of the ASIS questionnaire, the answers for each question were assigned a number 1 to 5. For each patient, the number was averaged per question, then averaged in respective categories of Signs, Impact, and Concerns. Average baseline ASIS-C scores were compared with ASIS-C scores after 12 weeks of treatment in all 10 patients as well as with ASIS-C scores in the patients that reached objective truncal or facial IGA success, defined as a >2-point decrease from baseline and IGA scores of clear (0) or almost clear (1).

Scores of the baseline answers of the 10 patients with moderate-to-severe acne were compared to the answers of the 10 acne-free control subjects to assess the % difference in the psychosocial difference between patients with moderate-to-severe acne and age-matched subjects with no acne. Comparisons in the % difference with the control subset were also made between all patients at week 12 and the patients who achieved IGA success at week 12, respectively.

## 3. Results

### 3.1. Patient Characteristics 

Ten patients participated in this study, with 5 males (50%) and 5 females (50%). Patient demographics are represented in Table 2. The patient’s average age was 16.3, with a range of 13–22. While all 10 patients had facial acne, an IGA score of 3 or 4 for facial acne was not an inclusion criterion. Sixty percent of patients achieved IGA success of the trunk after 12 weeks of sarecycline monotherapy, and 80% of patients achieved IGA success of the trunk or face. 

### 3.2. Control Characteristics

Ten acne-free control patients were analyzed in this pilot study, with 7 females (70%) and 3 males (30%). Age ranges are represented in Table 2.

### 3.3. Baseline ASIS-C Scores

At baseline in the 10 patients with moderate-to-severe truncal acne, the average ASIS-C score for the Signs domain was 2.7, for the Impact domain 2.85, and for the Concerns domain 2.56. In comparison, the control acne-free subjects had average scores of 1.52 for the Signs domain, 1.53 for the Impact domain, and 1.42 for the Concerns domain. Average Baseline scores of patients with truncal acne were higher than control acne-free subjects by 44% for Signs, 46% for Impact, and 45% for Concerns.

### 3.4. ASIS-C Scores after 12 Weeks of Sarecycline Therapy 

After 12 weeks of sarecycline monotherapy, average ASIS-C scores decreased from baseline ASIS-C scores by 4% in the Signs domain from 2.7 at baseline to 2.6, by 15% in the Impact domain from 2.9 to 2.4, and by 16% in the Concerns domain from 2.6 to 2.2. Average ASIS-C scores in the 60% of patients achieving IGA success decreased from baseline ASIS-C scores by 5% in the Signs domain from 2.6 to 2.5, by 20% in the Impact domain from 2.8 to 2.2, and by 19% in the Concerns domain from 2.4 to 2.0.

### 3.5. Percent Improvement in Sarecycline Responders

When comparing ASIS-C scores from patients achieving IGA success to control acne-free subjects, there were differences of 42% for Signs, 34% for Impact, and 29% for Concerns (Figure 1).

## 4. Discussion

Numerous studies have been performed in order to evaluate the impact of acne on health-related quality of life (HRQoL), but few have focused on the impact of truncal acne [10,11,12,13,14]. Psychosocial impacts include depression, low self-esteem, and suicidal ideation. In a cross-sectional, questionnaire-based study of 3775 participants, 14% reported having substantial acne (a lot and very much). Among those with “very much acne“ as compared to those with “no/little acne”, suicidal ideation was twice as frequently reported among girls and three times more frequently reported among boys [13]. 

Another qualitative study of 60 patients with facial and truncal acne and acne scars investigated the psychosocial burden through personification. The dominant theme that emerged included characterizations of acne as an intruder and an unwanted companion responsible for poor self-esteem and emotional impairment. It was demonstrated that the continuous burden of active acne starting from adolescence continues into adulthood besides clinically active lesions with scarring [14]. In a multi-country, population-based survey, it was demonstrated that facial and truncal acne together had a greater impact on HRQoL than facial acne alone. Importantly, more severe truncal acne was associated with an adverse impact on HRQoL irrespective of the severity of facial acne. This study utilized the dermatology life quality index (DLQI) and the Comprehensive Acne Quality of Life (CompAQ) [3].

PRO measures are an important tool in recognizing the patient’s perspective in the treatment of their acne. A recent systematic review from 2011 to 2021 evaluated how often PROs are utilized in the study of acne and rosacea. In this review of 206 randomized clinical trials (RCTs), 53% of rosacea and acne trials included at least 1 PRO. Only 7% of the RCTs analyzed included a PROM as a primary outcome [15]. This demonstrates that PROMs might be underutilized in the study of acne. 

ASIS was developed to provide a reliable and valid PROM that satisfied FDA criteria. The development process consisted of a literature review, concept elicitation interviews, item generation, and cognitive interviews. A literature review of other acne-specific PROMs revealed that the Acne Quality of Life Index (Acne-QOLI), Acne Quality of Life scale (AQOL), Acne-Specific Quality of Life Questionnaire (Acne-QoL), and the Dermatology-Specific Quality of Life Instrument for Acne (DSQL-Acne) were not in accordance with the FDA patient-reported outcome guidance. The framework was originally derived from the literature review and then refined by the concept of elicitation interviews. A 15-item draft questionnaire was created, which was finalized as a 17-item questionnaire following feedback from the cognitive interviews [5]. The questionnaire was divided into Signs and Impact domains to evaluate how the patients viewed their facial acne and how their acne impacted them, respectively. The Impact domain consists of two subscales, Emotional and Social.

Recently, 10 new questions were proposed for the Impact domain of ASIS that focus on patient Concerns (ASIS-C) and were included in our study. Answers to these questions were compared to the validated 17 questions in the original ASIS. In our reported study, the Concerns domain improved by 16% in all 10 patients and 19% in the patients who achieved IGA success. These results validate the 10 additional questions on Concern. Larger studies are needed for further validation of these additional questions.

Patient feedback was incorporated at each step of the development process of ASIS as the developers aimed to focus on patient-centered research. Items that patients identified as important during development were specific terminologies, such as the use of terms blackheads, whiteheads, scars, redness, dark marks, and overall acne. Other PROMs developed prior to ASIS did not include this. Similarly, symptoms addressed in existing PROMs such as pain, burning sensation, and soreness/tenderness were not identified as important by patients and, thus, not included in the ASIS. It is important to note that ASIS was designed to target patients with mild to moderate acne. Patients with severe cystic acne, acne conglobata, or any active or developing nodules were excluded from development [5]. 

Additionally, the developers chose different recall periods for the Signs and Impacts domain. For Signs, the recall period of “right now” was used to emphasize the symptoms that the patient is currently experiencing. For Impact, a recall period of 7 days was used to allow the patient time to process the impact of their current acne while still minimizing the time that a patient would have to rely on their memory [5]. Further, ASIS has been validated as a reliable PROM for facial acne. One hundred and fifty subjects completed baseline and follow-up assessments. Psychometric evaluation was performed using both traditional FDA psychometrics and new psychometric methods, such as the Rasch Measurement Theory (RMT). Both the Signs and Impact domains fulfilled traditional criteria and mostly satisfied Rasch criteria [6].

Few comparative studies evaluating the validity of PROMs have been performed. One study evaluating depression, anxiety, and life quality in acne patients treated with topical therapy versus isotretinoin used the DLQI, the Hospital anxiety and depression (HAD) scale, and the Beck Depression Inventory (BDI). Significant results were only found when comparing DLQI scores [16]. In a systematic review of 21 PROMs, only the CompAQ and the Acne-Q met the standards to be recommended for use in acne clinical studies. This study evaluated 10 acne-specific PROMs, 6 dermatology-specific PROMs, and 5 generic PROMs using the Consensus-Based Standards for the Selection of Health Measurement Instruments (COSMIN) criteria [17].

The 10 acne-specific PROMs included the Acne Disability Index (ADI), Acne Impact on Adult Daily Life (AI-ADL), ASIS, Acne-Q, AQOL, Acne-QoL, Acne-QOLI, Assessment of the Psychological and Social Effects of Acne (APSEA), Cardiff Acne Disability Index (CADI), and CompAQ. While CompAQ and the Acne-Q were the only PROMs to meet the COSMIN criteria in this study for content validity, it was found that all the PROMs studied were lacking a degree of content validity or other measurements. ASIS and ASIS-C should now be considered for measuring acne-associated HRQoL if additional evaluation of content validity is performed. ASIS was graded as moderate on Structural Validity, Internal Consistency, Reliability, and Construct Validity. Acne-Q was graded as low on Reliability and Construct Validity. The 6 dermatology-specific PROMs assessed included the DLQI, Children’s Dermatology Life Quality Index (CDLQI), Dermatology-Specific Quality of Life (DSQL), Oily Skin Self-Assessment Scale (OSSAS), Oily Skin Impact Scale (OSIS), and Skindex-29. The 5 generic PROMs analyzed were the UK Sickness Profile, EuroQol 5-Dimension, Short Form-36 (SF-36), Patient Benefit Index (PBI), and Patient-Reported Outcomes Measurement Information System–Anxiety. Each of the dermatology-specific and the general PROMs were graded as very low overall on content validity and may not be the most accurate tools to use in the assessment of acne. Overall, the PROMs remain difficult to assess given their qualitative nature. A major limitation of the study is that only what the individual PROM studies reported could be assessed. It is possible that all items in the COSMIN risk of bias checklists might not have been addressed due to limits on publication space or focus on other criteria [17].

Yoon et al. analyzed the content and phrasing of acne-specific PROMs with the conclusion that the PROMs have notable differences in the covered content, and PROM selection could potentially be guided depending on the clinical context. For example, if the primary focus is on psychological well-being, Acne-Q could be the most appropriate tool [18]. ASIS and ASIS-C are broader measures that encompass both the physical and emotional symptoms of facial acne, also considering the readability and length of the questionnaire [5].

Effective treatments for truncal acne are necessary to begin relieving the associated emotional impact. In line with PROMs, Auffret et al. developed the Truncal Acne Severity Scale (TRASS) as a patient-centered approach to validate treatment [19]. Seventeen current topical treatments for truncal acne include dapsone, azelaic acid, benzoyl peroxide, tretinoin, trifarotene, and clindamycin [20,21,22,23,24,25,26]. Systemic treatments include oral isotretinoin and tetracycline antibiotics [27,28]. Since few studies are designed to include truncal acne as an inclusion criterion, there is little objective data regarding the efficacy of these treatments [1]. Much less data exists on how these treatments affect the psychosocial aspect of truncal acne. One study of pooled data from 2 phase 3 trials on the efficacy and tolerability of dapsone 7.5% gel included ASIS as an endpoint and identified significant improvement across ASIS domains [29].

Sarecycline is a narrow-spectrum tetracycline-class antibiotic approved by the Food and Drug Administration (FDA) for moderate-to-severe acne vulgaris in patients 9 years of age and older [7]. Sarecycline demonstrates potent anti-inflammatory activity and Gram-positive activity against *Cutibacterium acnes*, a bacteria implicated in the pathogenesis of acne. Additionally, sarecycline has shown reduced activity against Gram-negative bacteria commonly found in the gut which may contribute to the favorable side effect profile and the low potential for antibiotic resistance [7]. Improvement in truncal acne with daily sarecycline treatment was observed in investigator global assessment (IGA) scores in phase 3 clinical trials, however, truncal acne was not an inclusion criterion [8,30]. While sarecycline has been reported as efficacious in the treatment of truncal acne, little data exists on the efficacy of sarecycline in relieving the psychosocial burden of truncal acne.

In this study, 60% of the patients treated with sarecycline achieved IGA success of the trunk, and 80% achieved IGA success of the trunk or face. Additionally, this study demonstrated success in improving scores in Signs, Impact, and Concern with sarecycline treatment in comparison with the control acne-free subjects.

It is of note that, since ASIS was developed for facial acne, the exact wording in the questions in the Signs domain may need to be modified by deleting the word “facial” in the questions in order for the questionnaire to be utilized more optimally for truncal acne studies. The inclusion of the word “facial” may have skewed the answers in the Signs domain for this study on truncal acne and diminished the true impact of the Signs domain and may explain the low impact on the Signs domain with sarecycline treatment. In order to reliably evaluate truncal acne, the word “face” should be edited to “skin”, for example. 

In this study, a large % difference is observed in the ASIS-C scores at baseline in the truncal acne patients compared to the control acne-free subjects; this illustrates the large psychosocial burden of truncal acne at baseline (Figure 1). After 12 weeks, the % difference in the ASIS-C scores from the control acne-free subjects decreases, even more so in those whose truncal acne cleared with sarecycline monotherapy; this suggests that ASIS-C scores, especially the Impact domain and the newly introduced Concern domain, may reflect the improvement in the psychosocial burden of acne. 

This pilot study is limited by the small sample size of 10 patients. A small sample size such as this could potentially lead to a type II error in which no real difference exists between the ASIS-C scores before and after treatment. The aim of this pilot study was to determine if there is a role for ASIS-C in the evaluation of the psychosocial burden of truncal acne. It was observed that the average answers decreased in each domain after 12 weeks of treatment for sarecycline. Despite the small sample size of this pilot study, our data suggest a need for further investigation of ASIS-C as a reliable PROM in truncal acne. 

## 5. Conclusions

Truncal acne is often unappreciated but carries a significant psychosocial burden, as the results in this study indicate. 

In this pilot study, the average ASIS answers decreased by 4% for the Signs domain, 15% for the Impact domain, and 16% in the Concerns domain after 12 weeks of treatment with sarecycline. ASIS and ASIS-C may be important PRO tools in evaluating psychosocial burden as well as improvement with successful acne treatment and would benefit from further validation in additional larger facial and truncal acne studies. 

## Figures and Tables

**Figure 1 antibiotics-12-00094-f001:**
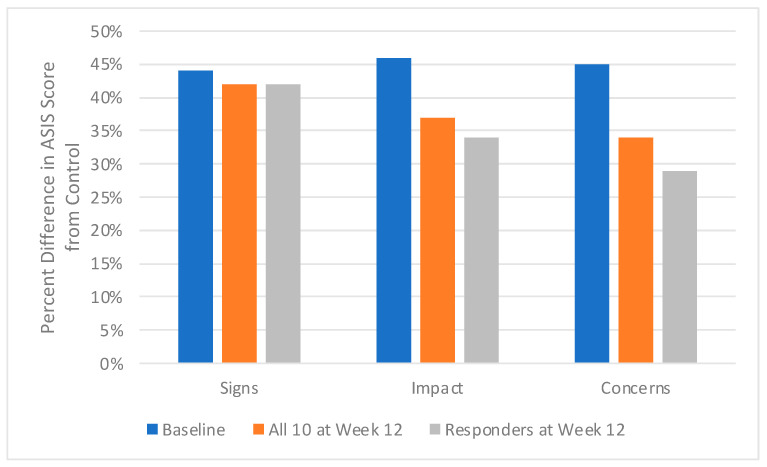
Acne Patients Before and After Oral Sarecycline Treatment.

**Table 1 antibiotics-12-00094-t001:** ASIS-C Questionnaire.

Questions	Answers
Signs	
How oily is your face right now?	Not at all to Very
How many pimples do you have on your face right now?	None to A lot
How many acne scars (holes or indents) do you have on your face right now?	None to A lot
How many scabs from acne do you have on your face right now?	None to A lot
How many dark marks from acne do you have on your face right now?	None to A lot
How many blackheads do you have on your face right now?	None to A lot
How many whiteheads do you have on your face right now?	None to A lot
How much redness do you have on your face right now?	None to A lot
Overall, how is the acne on your face right now?	None to A lot
Impact (Emotional)	
Over the past 7 days, rate how your face looked because of your acne.	Excellent to Bad
Over the past 7 days, how often did you feel sad because of the acne on your face?	Never to All of the time
Over the past 7 days, how often did you feel embarrassed because of the acne on your face?	Never to All of the time
Over the past 7 days, how often did you feel self-conscious because of the acne on your face?	Never to All of the time
Over the past 7 days, how often did you feel annoyed because of the acne on your face?	Never to All of the time
Over the past 7 days, how often did you feel not confident because of the acne on your face?	Never to All of the time
Impact (Social)	
Over the past 7 days, how often did you choose not to be around other people because of the acne on your face?	Never to All of the time
Over the past 7 days, how often did someone make bad comments about the acne on your face?	Never to All of the time
Concerns	
Over the last 7 days, how often has your acne made you feel angry (mad/sad)?	Never to All of the time
How worried are you about how long your acne will last and how bad it will get?	Not at all to Extremely
How often do you think about your acne?	Never to All of the time
Over the past 7 days, how worried have you been about your acne?	Not at all to Extremely
How often do you change, edit, or filter your social media photo or selfie because of your acne?	Never to All of the time
How often does acne impact your “in real-life” plans (IRL) (like dating or social engagements, playing sports, swimming, or hanging out)?	Never to All of the time
How often are you doing something to hide your acne (like mess with, squeeze/pop, or use makeup, concealer, hairstyle, or clothes to cover up)?	Never to All of the time
How often do you feel picked on or judged because of your acne?	Never to All of the time
How concerned are you that your acne will affect your ability to reach your future goals (in school or work) and be the best you can be?	Not at all to Extremely
Over the last 7 days, how often have you experienced worrying about or discomfort (itching/hurting) from acne?	Never to All of the time

**Table 2 antibiotics-12-00094-t002:** Patient Demographics for Sarecycline Arm and Control Arm.

Participant	Age	Sex
Sarecycline Arm		
1	14	Male
2	18	Male
3	13	Female
4	17	Male
5	14	Female
6	17	Male
7	15	Female
8	22	Female
9	18	Male
10	15	Female
Control Arm		
11	18–24	Female
12	18–24	Female
13	35–44	Male
14	18–24	Female
15	18–24	Female
16	18–24	Female
17	18–24	Female
18	25–34	Male
19	18–24	Male
20	18–24	Female

## Data Availability

Not applicable.

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
