# Peer review of "Effect of Sarecycline on the Acne Symptom and Impact Scale and Concerns in Moderate-to-Severe Truncal Acne in Open-Label Pilot Study"

_antibiotics, 2023, doi:10.3390/antibiotics12010094_

Round 1

Reviewer 1 Report

Dear authors,

I feel like the manuscript has severe flaws and needs to be polished in order to be accepted for publishing. 

The introduction is very, very, insufficient. It is necessary to explain the topic of the study better, to describe what is known and what is not known about this topic. 

The most important: The research goal and hypotheses are missing.The Material and method section is lacking.

What kind of study have you performed? Describe in detail the inclusion and exclusion .It is necessary better to connect your results with results from other studies and explain why.

The work is generally poorly conceived, despite the interesting and important topic.

I think this article has severe limitations and is not acceptable for publication in this journal.

Author Response

Reviewer 1:

Dear authors,

I feel like the manuscript has severe flaws and needs to be polished in order to be accepted for publishing.

The introduction is very, very, insufficient. It is necessary to explain the topic of the study better, to describe what is known and what is not known about this topic.

Thank you for taking the time to provide revisions on our article. The Introduction has been revised to include more background information regarding sarecycline, the drug in our study. We aimed to highlight what is known and what is not known.

The most important: The research goal and hypotheses are missing. The Material and method section is lacking.

What kind of study have you performed? Describe in detail the inclusion and exclusion .It is necessary better to connect your results with results from other studies and explain why.

The research goal and the hypothesis have been clarified within the introduction and throughout our manuscript, including in the Materials and Methods. The Materials and Methods section was expanded to include and clarify the inclusion and exclusion criteria as well as the type of study.

The work is generally poorly conceived, despite the interesting and important topic.

I think this article has severe limitations and is not acceptable for publication in this journal.

The aim of this pilot study was to evaluate if ASIS-C might play a role in the evaluation of the psychosocial impact of truncal acne, and we believe the results are important as to warrant further investigation.

Reviewer 2 Report

  1. Please write latin names in italic 
  2. Add study design to the title 
  3. Add study setting to the abstract 
  4. What kind of the introduction is This? Please rewrite, add investigated drug to the introduction 
  5. Table 2 should be deleted and Data presented in the manuscript 
  6. The study needs to be conducted on a larger sample

Author Response

Reviewer 2:

Please write latin names in italic

Add study design to the title

Add study setting to the abstract

Thank you for taking the time to provide revisions on our article. We made the suggested edits of italicizing latin names, adding study design to title, and adding the study setting to the abstract.

What kind of the introduction is This? Please rewrite, add investigated drug to the introduction

The Introduction has been revised to include more background information regarding sarecycline, the drug in our study. Additionally, the research goal and the hypothesis have been clarified within the introduction and throughout our manuscript.

Table 2 should be deleted and Data presented in the manuscript

Table 2 is not meant to present data, but rather demonstrate the contents of the ASIS-C questionnaire that the subjects received. More detail was added to the Methods section to clarify this.

The study needs to be conducted on a larger sample

This pilot study is limited by the small sample size of 10 patients. The aim of this pilot study was to evaluate if ASIS-C might play a role in the evaluation of the psychosocial impact of truncal acne, and we believe the results are important as to warrant further investigation.

Reviewer 3 Report

The study is very simple and no novelty was found. The sample size is very low, so what is the reason of very low sample size.

Methodology is unclear, which type of study, design of study, how to calculate the sample size, how to validate the questions, the added 10 questions, what is the criteria to add these questions.

Overall many unseen questions are in mind when read the methodology.

What is criteria of moderate to severe acne?

what is location of study?

no statistical analysis was done.

Suggested to add these all in paper and re-submit again.

On what criteria patients were selected. 

Author Response

Reviewer 3:

The study is very simple and no novelty was found. The sample size is very low, so what is the reason of very low sample size.

Thank you for taking the time to provide revisions on our article. This pilot study is limited by the small sample size of 10 patients. The aim of this pilot study was to evaluate if ASIS-C might play a role in the evaluation of the psychosocial impact of truncal acne, and we believe the results are important as to warrant further investigation.

Methodology is unclear, which type of study, design of study, how to calculate the sample size, how to validate the questions, the added 10 questions, what is the criteria to add these questions.

Overall many unseen questions are in mind when read the methodology.

What is criteria of moderate to severe acne?

what is location of study?

no statistical analysis was done.

Suggested to add these all in paper and re-submit again.

On what criteria patients were selected.

The Materials and Methods section was expanded to include and clarify the inclusion and exclusion criteria, the type of study, the study design, criteria of moderate to severe acne, and location of the study. No statistical analysis was done as this is a qualitative study evaluating the questionnaire before and after treatment.

Reviewer 4 Report

The manuscript actually mostly deals with the evaluation of acne using various tools, and your study seems to be secondary to that. The chapter Material and Methods placed after the Discussion chapter does not ease that impression. That is why I would consider that the title does not really describe the article's content, and it probably would be better to change it.
The study as it is presented is merely a micro-study, with actually only 10 patients involved and with rather modest responses (the average ASIS answers decreased by 4% for the Signs domain, 15% for 237 in the Impact domain, and 16% in the Concerns domain after 12 weeks of treatment with 238 sarecycline). Probably it would be more significant if you would have enrolled in a greater number of subjects, offering some statistical relevance to your work.

Author Response

Reviewer 4:

The manuscript actually mostly deals with the evaluation of acne using various tools, and your study seems to be secondary to that. The chapter Material and Methods placed after the Discussion chapter does not ease that impression. That is why I would consider that the title does not really describe the article's content, and it probably would be better to change it.

Thank you for taking the time to provide revisions on our article. We agree with your suggestion and the title of the manuscript has been changed to “Effect of Sarecycline on the Acne Symptom and Impact Scale and Concerns in Moderate-to-Severe Truncal Acne in Open-Label Pilot Study” to better represent the aim of our study.

The study as it is presented is merely a micro-study, with actually only 10 patients involved and with rather modest responses (the average ASIS answers decreased by 4% for the Signs domain, 15% for 237 in the Impact domain, and 16% in the Concerns domain after 12 weeks of treatment with 238 sarecycline). Probably it would be more significant if you would have enrolled in a greater number of subjects, offering some statistical relevance to your work.

The Material and Methods section has been revised and moved to after the introduction. This pilot study is limited by the small sample size of 10 patients. The aim of this pilot study was to evaluate if ASIS-C might play a role in the evaluation of the psychosocial impact of truncal acne, and we believe the results are important as to warrant further investigation.

Round 2

Reviewer 1 Report

Dear authors,

The manuscript has improved a lot after the revision. 

Reviewer 2 Report

Nicely Done. Congrats 

Reviewer 3 Report

The modifications may be accepted.

Reviewer 4 Report

-